# Equivalent Shell Model of Elastic Gridshells Including the Effect of the Geometric Curvature

**Maria Luisa Regalo †, Stefano Gabriele †** **, Valerio Varano †** **and Ginevra Salerno \*,†**

LIMeS—Applied Mathematics and Structural Mechanics Laboratory—Department of Architecture,
University Roma Tre, Via della Madonna dei Monti, 40-00184 Rome, Italy;
marialuisa.regalo@uniroma3.it (M.L.R.); stefano.gabriele@uniroma3.it (S.G.); valerio.varano@uniroma3.it (V.V.)
\* Correspondence: ginevra.salerno@uniroma3.it
† These authors contributed equally to this work.

**Abstract:** In this work, an equivalent continuum of a barrel gridshell is introduced. Constitutive identification procedures based on periodic homogenization are provided in the literature for this purpose, based on a flat Representative Element Volume (REV), notwithstanding that the geometry of the structures concerned is curved. Therefore, the novelty of the present study is the selection of a curved REV to obtain the equivalent elastic constants. The numerical validation of the identification procedure is made comparing gridshell response to that of the equivalent shell under homogeneous load conditions. Finally, in order to highlight the effect of the curved geometry on the constitutive law of the continuum, the response of the proposed model is also compared to that of a continuum obtained from a flat REV.

**Keywords:** elastic gridshells; shell model; constitutive identification; geometric curvature



## 1. Introduction

A gridshell has a doubly curved geometry as a shell; however, its material is laid out in a grid pattern and that is the reason why it is also called a lattice shell or reticulated shell.

This kind of structure may be analysed as systems of beam elements, from now on referred to as "beam lattice" models, through extensions of the ordinary theories for structural frames, or alternatively they may be analysed as continua.

Although the propagation of highly efficient algorithmic tools for the analysis of the beam lattices is constantly growing (FEM and DEM software [1,2] or Cad tools endowed with Isogeometric analysis [3]), the deeper understanding of their mechanics is increasingly difficult to achieve. As a result, the designers are not protected against errors originating from the inability to check the solution of automatic computations, especially at the initial phases.

Analysing the static behaviour of beam lattices in parallel with that of their equivalent continua may be a solution to this problem.

In the second half of the twentieth century, the attraction for this approach led to defining different methods for establishing beam lattice-continuum relations. One of them is the *equivalent stiffness method*, conceived by Wright in [4]; the procedure provides the properties of the material and the effective thickness of the continuum model as a function of the geometry and the mechanical properties of the beam lattice shell, paying attention only on the membrane behaviour. Another method, called *split rigidity method* and introduced by K. P. Buchert in [5], is based on two different rigidities, one for the membrane deformations and the second for the bending ones. A further method is the *orthotropic equivalent continuum method* introduced in [6], based on an orthotropic constitutive law to find the rigidities of the equivalent shell.

The same results found in [6] are obtained by the authors of the present work in [7,8], where the orthotropic equivalent continuum, used to study the consequences of different orientations of gridshell laths, is obtained by homogenization using a flat REV.

One of the first validations of the equivalent continuum defined by the equivalent stiffness method has been carried out in [9]. In particular, the study compares the accuracy of the continuum for describing the buckling phenomenon and it concludes that the equivalent shell model could be employed only as preliminary tool to estimate the critical buckling load of a lattice shell.

Defining an equivalent continuum together with its validation is the dominant paradigm nowadays, as also shown in other application fields, such as periodic brickwork [10,11] and, recently, in nano-structures [12,13].

It is evident that the continuum model provides a less accurate description of the behavior of a gridshell than a beam lattice model. In this sense, using the denominations *coarse* and *fine*, respectively, to indicate the two models is justified. However, even if the continuum (coarse) model is less accurate than the beam lattice (fine) model, the former can be used if the loss of accuracy is compensated for by a considerable increase in usability [14].

Therefore, the continuum model is not considered as an alternative to the beam lattice one, but rather as a complementary tool; it is highly preferred in the first phase of a design, when the effectiveness of the global geometry of the gridshell has to be evaluated, an overall view of the distribution of internal forces is needed, the topologies of the grid and its orientation are still to be decided, and approximate values of bars forces are looked for.

The aim of this work is to provide a proper definition of the equivalent stiffness describing the transition from the *fine* model to the *coarse* one, that is, from the beam lattice model to the continuum shell. The aforementioned methods are addressed to beam lattice shells approximating shell surfaces within a framework of relatively short linear pieces; therefore, they do not consider the influence of the curved geometry. However, the equivalent continuum defined in [4] has also been used in the initial design phase of the Mannheim Multihalle, a timber gridshell assembled from a lattice of curved continuous laths [15]. That was justified since the ratio of the mesh size over to the sgrid span is sufficiently small.

However, how small does that ratio have to be so that the equivalent model is reliable? The present authors look for an answer to this question introducing, in the context of linearized elasticity, a new equivalent continuum for a simple barrel gridshell, deriving the constitutive coefficients through an identification procedure taking into account the effects of the curved geometry. Generally speaking, the identification of curved structures runs into complexities that depend on the value of the Gaussian curvature. Addressing the problem of curvature for the first time, having worked on the same theme in the past by using a flat REV, what we are doing now is to take into account the curvature on an extremely simple case, characterized by null Gaussian curvature. The geometric choice is accompanied by a further simplification, which limits the analysis to the coupling between axial and bending stresses, focusing attention only on symmetrical homogeneous states. Obviously, when we used the flat REV these coupling effects were neglected a priori.

To recap, we identify only some constitutive coefficients, with the ambition, on the one hand, to verify the effects of unidirectional curvature and, on the other hand, to focus on the aforementioned coupling. This identification is the subject of Section 2.1. Then, the numerical validation of the identification procedure is made comparing beam lattice response to that of the equivalent shell under homogeneous load conditions in Section 3.1. In order to show the actual influence of the curved geometry in the continuum formulation, the obtained equivalent continuum is finally compared, in Section 3.2, with the one proposed in [6], which neglects the curvature of the reference surfaces. The conclusions follow.

## 2. Materials and Methods

### *2.1. The Constitutive Identification*

#### 2.1.1. Methodological Premise

To determine the elastic coefficients of the continuum model equivalent to the beam lattice shell, a continualization procedure of a periodic system is used, which follows the lines of the homogenization methods of heterogeneous continuous models.

This procedure starts from the selection of an elementary reference volume (REV) in the fine model (the beam lattice), determined by the periodicity of the system, and identifies in the coarse model (the continuum) a part that occupies the same space region of the REV.

At this point, a point-wise correspondence is defined between the internal forces in the two models, coarse and fine ones, though a localization operator and, then, equality is imposed between the respective strain energies, expressed in terms of complementary energy. The correspondence between fine and coarse inner forces allows the strain energy of the fine model be expressed in terms of the continuous inner forces; the elastic coefficients of the continuum model are finally obtained by successive differentiations.

The determination of the strain energy of the fine REV passes through the resolution of a series of hyperstatic elastic problems, to which external forces of the bulk type and force boundary conditions are imposed. These problems are solved by the force method, after the imposition of constraints sufficient to remove the rigid motion. The simplicity of the beam lattice model allows the exploitation of elastic solutions in the literature ([16]), which provide the displacement values in analytical terms. For reasons of simplicity, therefore, the evaluation of the strain energy of the fine model is carried out by estimating the external work, using the principle of virtual works.

#### 2.1.2. The Fine and the Coarse Models: The REV

The beam lattice REV of a barrel gridshell is shown in Figure 1 on the left. The selected gridshell has a quadrangular mesh, whose members are modelled as Bernoulli beams (see [16] for equations). In particular, members can be divided into two groups: the longitudinal ones, that are straight beams, and the transversal ones, that are circular arches. Arches and beams are connected through panthographic joints at the intersections, in order to guarantee the continuity of the members. Finally, the quadrangular grid is triangulated by truss elements to provide in-plane shear stiffness.

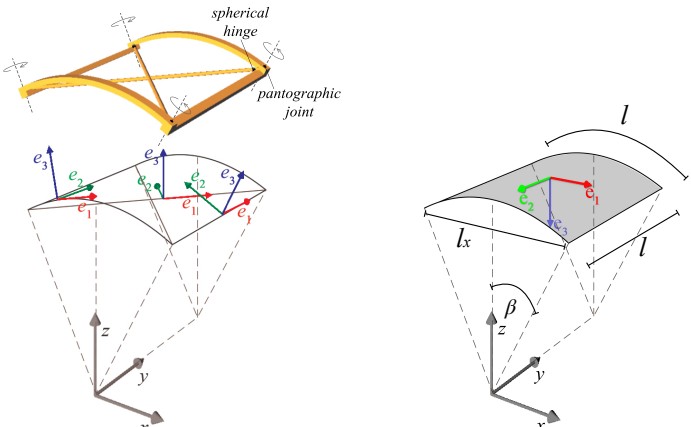

**Figure 1.** (**Left**) The fine geometry of the REV, together with local and global axes. The rotation axes for pantographic joints are shown. (**Right**) The corresponding continuum region, together with local, global axes and dimensions.

The model is immersed in a three-dimensional space, described by a global coordinate system $xyz$. In addition, each member is endowed with the own local coordinate system $e_1e_2e_3$; in particular, $e_1$ is tangent to the centroid axis of the member and $e_2$ and $e_3$ are the

principal axes of inertia of the cross-section. Global and local coordinate systems of any member of the gridshell module are shown in Figure 1.

For the coarse model, a Donnell shell is selected, whose equations are described in [17] and recalled later.

Even the continuum model is endowed with an own local coordinate system $e_1 e_2 e_3$; the first two local axes belong to tangent plane to the surface while the third one is perpendicular to it (Figure 1). Local axis $e_1$ is tangent to the curved generator of the cylindrical shell, while $e_2$ is parallel to the straight one.

The identification procedure starts from the choice of a Representative Elementary Volume (REV) in the beam lattice model and the selection of a region in the continuum model of the same area as that occupied by the REV. Here, the module of the beam lattice shown in Figure 1 on the left is chosen as REV. The geometry of the REV is defined as having the length of the longitudinal straight beams and of the arches equal to $l$. Moreover, the arches are circular with radius $R$, and center angle $2\beta = \frac{l}{R}$.

The same figure shows the corresponding continuum region on the right.

To recap, the REV is composed of six elements: two curved beams, two straight beams and two pin-jointed bars, i.e., the diagonals. In addition, all the beams have been considered with half the area of their cross-sections and half the second moment of area, for reasons of periodicity, since they are common to two contiguous modules.

### 2.1.3. The Localization Operator

The objective is defining the constitutive law which describes membrane and bending behaviour of the continuum model, so that it is capable of providing strain states equivalent to the beam lattice model ones, under equivalent stress states. This is done by equalizing the internal work of the fine model to the internal work of the coarse one, obtained from the equivalent stress states.

Figure 2 shows the uniform membrane and bending stresses acting on the sections of normal vectors $e_1$ and $e_2$, which we collect in the vector $\mathbf{s}^T = \{N_{11}, N_{22}, M_{11}, M_{22}\}^T$.

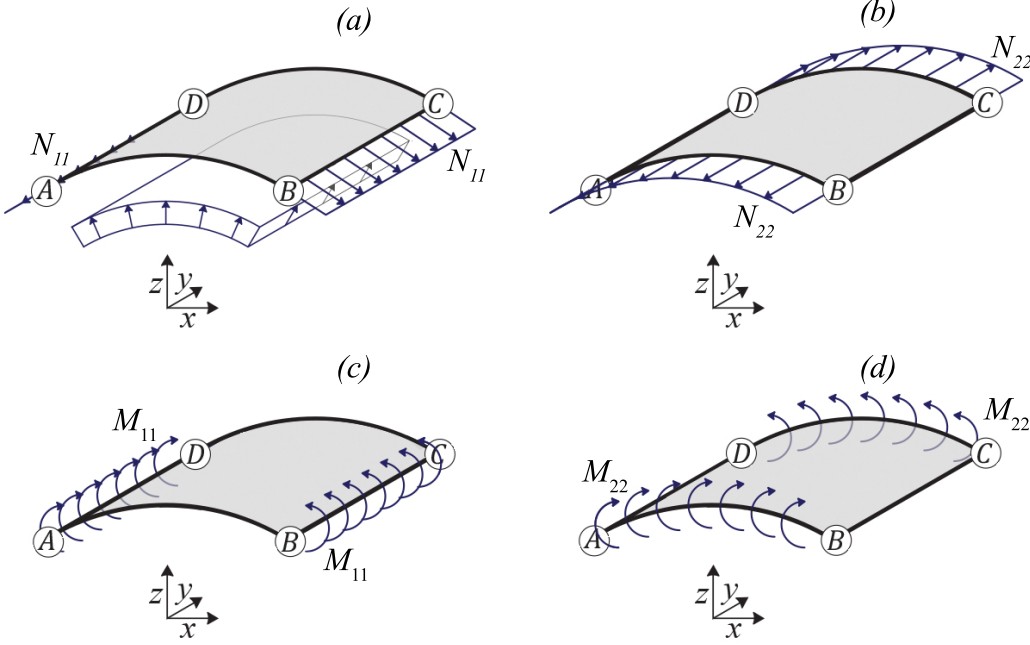

**Figure 2.** Internal actions in the continuum region corresponding to the REV for the four uniform states: (**a**) $N_{11}$ membrane axial action along $e_1$ (**b**) $N_{22}$ membrane axial action along $e_2$ (**c**) $M_{11}$ bending action about $e_2$ (**d**) $M_{22}$ bending action about $e_1$

Likewise, membrane and bending strains along the same directions are collected in the vector $\mathbf{e}^T = \{\epsilon_1, \epsilon_2, \chi_1, \chi_2\}^T$. Their relationship is supposed to be linear, that is:

$$\mathbf{e} = \mathbf{Ds}, \tag{1}$$

where $\mathbf{D}$ is the flexibility matrix, whose coefficients are the unknowns of the identification procedure. In very general terms, it is written as follows:

$$\mathbf{D} = \begin{bmatrix} D_{11} & D_{12} & D_{13} & D_{14} \\ & D_{22} & D_{23} & D_{24} \\ & & D_{33} & D_{34} \\ sym & & & D_{44} \end{bmatrix} \tag{2}$$

At this point, the continuum stress states, depicted in Figure 2, are made equivalent to the stress states of the beam lattice model and shown in Figure 3.

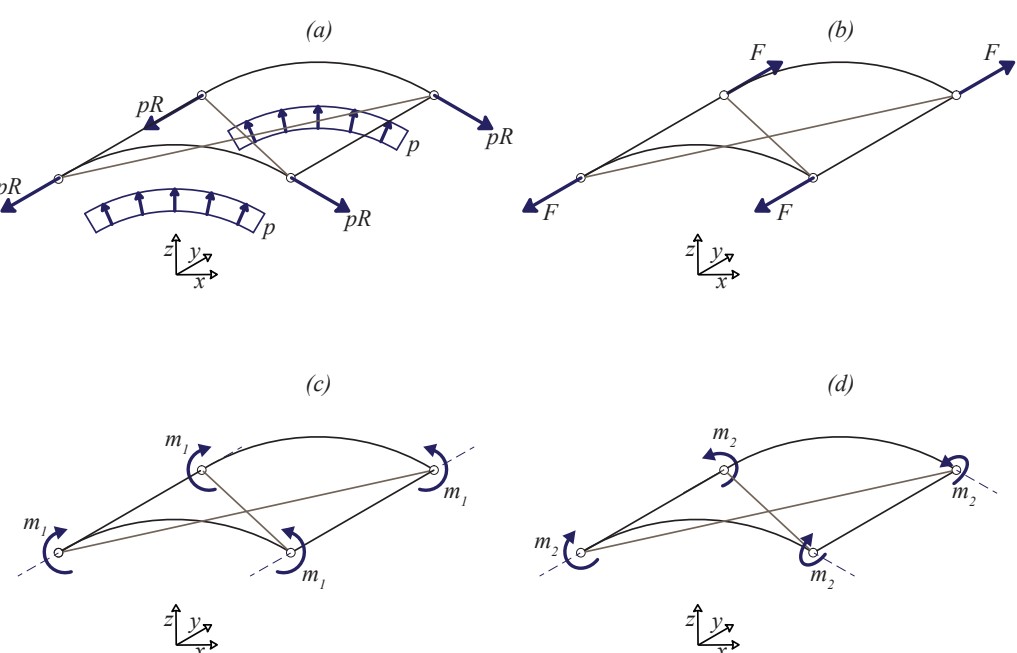

**Figure 3.** The four elastic problems induced to the REV by the localization operator, starting from the four uniform states of the continuum model. Refers to Equation (3) for definitions of boundary actions of (**a**), (**b**), (**c**) and (**d**) respectively.

By means of the following equations:

$$p = N_{11}l/2R, \quad F = N_{22}l/2, \quad m_1 = M_{11}l/2, \quad m_2 = M_{22}l/2, \tag{3}$$

where the radial pressure $p$ on the arches corresponds to the radial bulk force $p/l$ in the coarse model. This equation, often referred to as localization operator in standard homogenization methods, induces four elastic problems on the REV, shown in Figure 3: these are traction problems, due to the force identification procedure used up to now.

The four load conditions are the following:

1. Uniform pressure $p$ applied to the arches in the plane $e_2, e_3$, in equilibrium with the tensile forces $pR$ at the end sections of the arches (Figure 3a);
2. Tensile forces $F$ applied at the end sections of the straight beams (Figure 3b);
3. Couples $m_1$ applied at the end sections of the arches (Figure 3c);
4. Couples $m_2$ applied at the end sections of the straight beams (Figure 3d).

To any traction problem kinematic constraints are added to eliminate any rigid motions and for reasons of periodicity. It is hardly worth pointing out that the end sections of arches

are located on planes of symmetry of the fine model, on which all antisymmetric quantities, such as rotations, axial displacements and shear forces cancel out, for obvious reasons. Another plane of symmetry for the REV passes through the two center points of the straight beam axes (see Figure 4). These geometric conditions impose specific restrictions on the displacements, which are defined in Section 3, where, as those problems statically indeterminate, the force method is used to solve them.

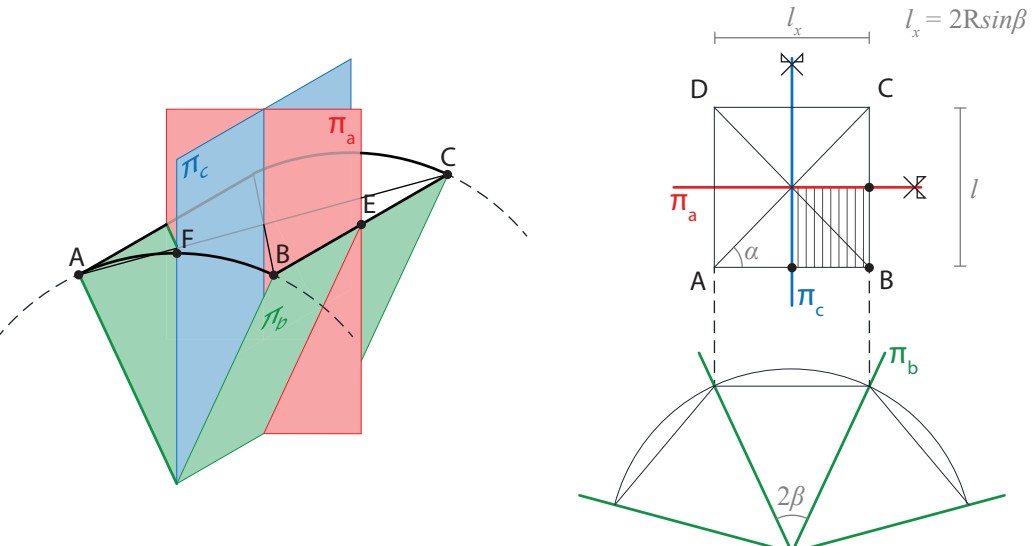

**Figure 4.** Symmetries of the problems. (**Left**) 3D view. (**Right**) Top and Front view. The symmetries of the REV (w.r.t. $\pi_a$ and $\pi_c$) together with periodicity conditions (w.r.t. $\pi_b$) enable to impose the boundary conditions required to eliminate rigid displacements of the REV.

### 2.1.4. Work Equality and the Elastic Coefficients

The solution of the elastic problems defined in Section 2.1.3 is presented in some detail in Section 2.2 and allows the strain energy of the fine model be defined. As mentioned in the methodological premise, in this case it is easier to determine the strain energy through the work done by the external forces on the corresponding displacements. External forces are those defined by the localization operator Equation (3). The only bulk force is the radial pressure, which acts in the plane of the arches and spends work for the radial displacement $w$ of the axis line, while the remaining forces and torques spend work on the displacements and the rotations of the four nodes $A$, $B$, $C$ and $D$. Therefore, the external work takes the following expression:

$$W^{ef} = 2\int_{-l/2}^{l/2} p\, w(s)ds + \sum_{k=A}^{D} pRu_k + \sum_{k=A}^{D} Fv_k + \sum_{k=A}^{D} m_1\phi_k + \sum_{k=A}^{D} m_2\theta_k, \tag{4}$$

where $u$ is the displacement tangent to the axis line of the arches' end sections, $v$ the axial displacement of the straight beams' end sections, and $\phi$ and $\theta$ are the components of nodes' rotations around the axis $e_2$ of the arch and the axis $e_2$ of the beam, respectively.

Symmetry reasons impose the equality in norm of the homologue displacement components of the four edge nodes and transform the Equation (4) into the following:

$$W^{ef} = 2\int_{-l/2}^{l/2} p\, w(s)ds + 4pRu + 4Fv + 4m_1\phi + 4m_2\theta \tag{5}$$

which in turn, by replacing the Equation (3) in it, is modified as follows:

$$W^{ef} = \frac{N_{11}l}{R}\int_{-l/2}^{l/2} w(s)ds + 2N_{11}ul + 2N_{22}vl + 2M_{11}\phi l + 2M_{22}\theta l \tag{6}$$

At this point the solutions of the elastic problems, defined in Section 2.1.3 and which are being treated extensively in Section 2.2, can be used to simplify Equation (6). First of all, strict periodicity reasons impose $u = 0$, canceling the second addend of Equation (6).

In addition, anticipating fragments of the solution, we have:

$$
\begin{aligned}
w(s) &= N_{11}\hat{w}_1(s) + N_{22}\hat{w}_2(s) + M_{11}\hat{w}_3(s) \\
v &= N_{11}\hat{v}_1 + N_{22}\hat{v}_2 + M_{11}\hat{v}_3 \\
\phi &= N_{11}\hat{\phi}_1 + N_{22}\hat{\phi}_2 + M_{11}\hat{\phi}_3 \\
\theta &= M_{22}\hat{\theta}_4
\end{aligned}
\tag{7}
$$

with obvious meaning of the symbols used. Thus, Equation (6) transforms into:

$$
\begin{aligned}
W^{ef} =& N_{11}\int_{-l/2}^{l/2}\left(N_{11}\hat{w}_1(s) + N_{22}\hat{w}_2(s) + M_{11}\hat{w}_3(s)\right)ds\frac{l}{R} + \\
& 2N_{22}\left(N_{11}\hat{v}_1 + N_{22}\hat{v}_2 M_{11}\hat{v}_3\right)l + \\
& 2M_{11}\left(N_{11}\hat{\phi}_1 N_{22}\hat{\phi}_2 + M_{11}\hat{\phi}_3\right)l + \\
& 2M_{22}^2\hat{\theta}_4 l
\end{aligned}
\tag{8}
$$

which, after some algebra, turns out to be a quadratic form of the coarse stress state $\mathbf{s}^T = \{N_{11}, N_{22}, M_{11}, M_{22}\}^T$:

$$
\begin{aligned}
W^{ef} =& N_{11}^2\int_{-l/2}^{l/2}\hat{w}_1(s)ds\frac{l}{R} + \\
& N_{11}N_{22}\left(\int_{-l/2}^{l/2}\hat{w}_2(s)ds\frac{1}{R} + 2\hat{v}_1\right)l + \\
& 2N_{22}^2\hat{v}_2 l + \\
& N_{11}M_{11}\left(\int_{-l/2}^{l/2}\hat{w}_3(s)ds\frac{1}{R}\right)l + \\
& 2N_{22}M_{11}\left(\hat{\phi}_2 + \hat{v}_3\right)l + \\
& 2M_{11}^2\hat{\phi}_3 l + \\
& 2M_{22}^2\hat{\theta}_4 l
\end{aligned}
\tag{9}
$$

The external work of the fine model $W^{ef}$ is ready to be equated with the internal work of the coarse model, given by

$$
W^{ic} = l^2\mathbf{e}\cdot\mathbf{s} = l^2\mathbf{D}\mathbf{s}\cdot\mathbf{s}
\tag{10}
$$

where $\mathbf{e}$, $\mathbf{s}$ and $\mathbf{D}$ are defined in Section 2.1.3. Taking into account Equations (9) and (10) assumes the following form:

$$
\begin{aligned}
W^{ic} = l^2(& D_{11}N_{11}^2 + 2D_{12}N_{11}N_{22} + D_{22}N_{22}^2 + \\
& 2D_{13}N_{11}M_{11} + 2D_{23}N_{22}M_{11} + \\
& D_{33}M_{11}^2 + D_{44}M_{22}^2)
\end{aligned}
\tag{11}
$$

with the elastic constants $D_{11}, D_{12}, D_{13}, D_{22}, D_{23}, D_{33}, D_{44}$ given by:

$$D_{11} = \frac{1}{lR} \int_{-l/2}^{l/2} \hat{w}_1(s) ds \tag{12}$$

$$D_{12} = \frac{1}{2lR} \int_{-l/2}^{l/2} \hat{w}_2(s) ds + \frac{1}{l} \hat{v}_1 \tag{13}$$

$$D_{22} = \frac{2\hat{v}_2}{l} \tag{14}$$

$$D_{13} = \frac{1}{2lR} \int_{-l/2}^{l/2} \hat{w}_3(s) ds + \frac{\hat{\phi}_1}{l} \tag{15}$$

$$D_{23} = \frac{\hat{\phi}_1}{l} + \frac{\hat{v}_3}{l} \tag{16}$$

$$D_{33} = \frac{2\hat{\phi}_3}{l} \tag{17}$$

$$D_{44} = \frac{2\hat{\theta}_4}{l} \tag{18}$$

It is hardly worth noting that the constitutive coefficients in Equations (12)–(18) have different physical dimensions: $[D_{11}] = [D_{22}] = [D_{12}] = [L\,F^{-1}]$ ,$[D_{13}] = [D_{14}] = [D_{23}] = [D_{24}] = [F^{-1}]$ , $[D_{33}] = [D_{34}] = [D_{44}] = [L^2\,F^{-1}]$. On the other hand, it is more interesting to note that the quantities $\hat{a}$ of the linear combinations of Equations (7) do not have the physical dimension of displacements and rotations, but of derivatives of generalized displacements with respect to the internal forces of the coarse model, $N_{11}, \dots, M_{22}$, and, therefore, they contain the constitutive coefficients of the fine model, as is going to be shown later. Moreover, this is the reason for some apparent contradictions, like the one in Equation (16).

In addition, the mechanical couplings that can be deduced from Equations (13), (15) and (16) are worth to be underlined. The coefficient $D_{12}$, coupling $N_{11}$ and $N_{22}$, takes into account the Poisson effect in the membrane behavior, while the coefficients $D_{13}$ and $D_{23}$, coupling $N_{11}$ and $M_{11}$, and $N_{22}$ and $M_{11}$, represent the answer to one of the questions raised by this paper. Furthermore, if in the plane of the arch the coupling measured by $D_{13}$ is practically discounted, the coupling in the perpendicular plane, highlighted by $D_{23}$, is much less obvious.

This last coefficient represents the fact that when $M_{11}$, while bending the arch in its plane, it lengthens its axis line, at the same time it also causes the elongation of the diagonals which, in turn, compress the longitudinal beams. On the contrary, the bending moment $M_{22}$ produces effects that remain confined in its plane, because since it does not lengthen the longitudinal beams, it does not lengthen the diagonals either, which are the coupling element of the two perpendicular planes.

The detailed expression of the elastic constants is given in Section 2.4.

A final consideration is work making. The elastic constants determined above make Donell's shell model equivalent to the barrel beam lattice under examination. Having used a force-based approach, the stress states of the two models correspond point-wise via the localization equation Equation (3), while the deformation of the coarse model provides only an average measure of the deformations of the fine model. In fact, it is well known that the deformed shape of the fine model is characterized by local phenomena, which are lost in the coarse model.

For the determination of the displacements of the fine model, starting from the solutions of elastic problems of the equivalent Donnell shell, Equation (7) is used.

### 2.2. The Solutions of the Fine Model

The analytical solution of the four elastic problems defined on the REV can be easily found by exploiting the symmetries shown in Figure 4. These are due to the symmetry

of the REV, with respect to the vertical planes $\pi^a$, $\pi^c$ together with the global cylindrical symmetry that makes the circumferential components of the nodes displacements null.

In particular the symmetries allow to solve the whole problem by:

1. Solving the sub-problem consisting of one half arch, one half diagonal and one half beam converging in the node B (see the rectangular hatched area in Figure 4);
2. Assuming that the displacement of the node B belong to the plane $\pi^b$.

### 2.3. The Solution Method

The symmetry and periodicity conditions allow each of the four 3D problems be solved as a linear combination of plane 1D problems. The 3D structure is statically un-determined with one degree of redundancy. The Force Method is used, where the axial force in the diagonal is assumed as redundant reaction. Each problem is then solved as the superposition of five elastic 1D plane statically determined sub-problems, by making use of the following scheme, where the index $i = 1, \ldots, 4$ represents the quantities referred to the $i$th problem, while the apices $a, b, d$ are referred to (arch, beam, diagonal), respectively:

- The axial force in the diagonal is assumed as redundant reaction $X_i$ (see Figure 5).
- The redundant reaction $X_i$ is projected onto the vertical plane $\pi^a$ containing the arch and in the plane $\pi^b$ containing the beam, by obtaining the components $X_i^a = X_i \cos(\alpha)$ and $X_i^b = X_i \sin(\alpha)$, respectively, where $\alpha$ is the angle formed between the diagonal and the plane $\pi^a$ in the reference configuration.
- Five statically determined elastic traction sub-problems are solved:
    - $S_i^{ao}$: The arch subject to the external loads (if present);
    - $S_i^{ax}$: The arch subject to $X_i^a$;
    - $S_i^{bo}$: The beam subject to the external loads (if present);
    - $S_i^{bx}$: The beam subject to $X_i^b$;
    - $S_i^{dx}$: The diagonal subject to $X_i$.
- For each sub-problem the following six constraints are assumed on the plane displacements of the half arch and half beam:

$$u_i^a(0) = 0, \phi_i^a(0) = 0, u_i^a(l/2) = 0, u_i^b(0) = 0, w_i^b(-l/2) = w_i^a(l/2), \theta_i^b(0) = 0$$

- The plane displacements $\{u_i^a(s), w_i^a(s), \phi_i^a(s)\}^T$ of the arch are obtained as sum of the results of the problems $S_i^{ao}$ and $S_i^{ax}$, the plane displacements $\{u_i^b(s), w_i^b(s), \phi_i^b(s)\}^T$ of the beam are obtained as sum of the results of the problems $S_i^{bo}$ and $S_i^{bx}$, while the displacements $u_i^d(s)$ of the diagonal comes from the solution of the problem $S_i^{dx}$. All the displacements are written in terms of $X_i$.
- The compatibility condition between the axial displacement of the diagonal and the projection of the axial displacements of arch and beam on the *x-y* plane is used in order to find the value of $X_i$.

$$u_i^d(l/2) = \cos(\alpha)(u_i^a(l/2)\cos(l/2R) + w_i^a(l/2)\sin(l/2R)) + \sin(\alpha)u_i^b(l/2) \quad (19)$$

- The whole displacement field for each element is finally written in terms of the external load only.

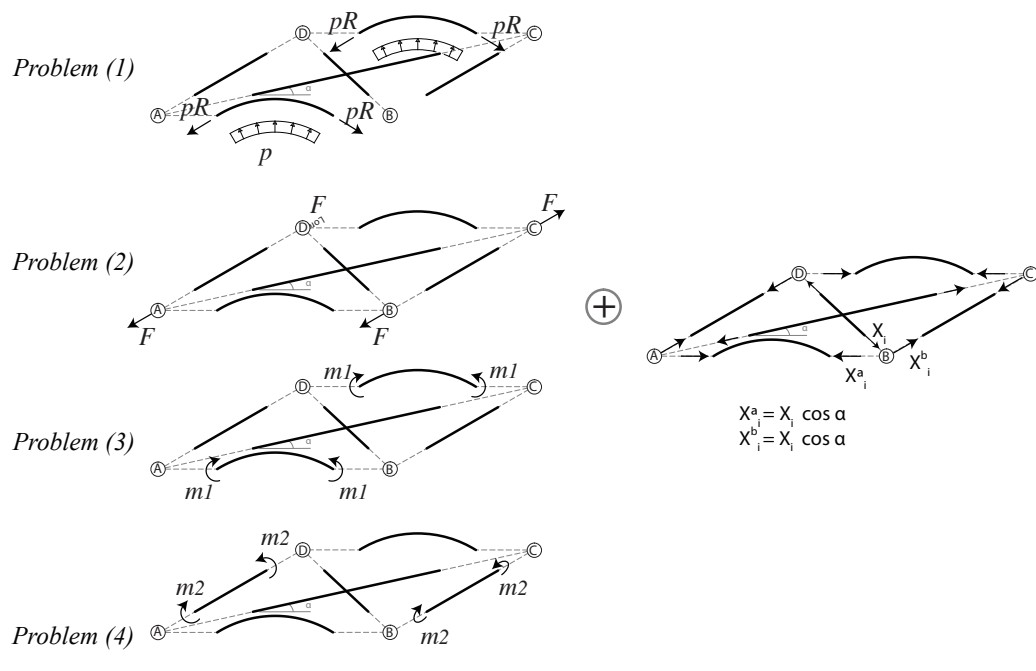

PRIMARY STRUCTURE          REDUNDANT X

**Figure 5.** Scheme of the Force Method used to solve the four fine problems. (**Left**) The four different primary structures, one for each problem. (**Right**) The auxiliary structure, characterized by the redundant reaction $X$.

### 2.3.1. The Equations of the Three Plane Problems

As well known, the elastic problems for the arch, beam and diagonal are defined by the following systems of equations:

Elastic problem for the arch:

$$
\begin{cases}
\varepsilon_i^a(s) &= (u_i^a)'(s) + \frac{w_i^a(s)}{R} \\
\gamma_i^a(s) &= (w_i^a)'(s) - \frac{u_i^a(s)}{R} - \phi_i^a(s) \\
\chi_i^a(s) &= (\phi_i^a)'(s)
\end{cases}
\begin{cases}
N_i^a(s) &= EA_b \varepsilon_i^a(s) \\
\gamma_i^a(s) &= 0 \\
M_i^a(s) &= EJ_b \chi_i^a(s)
\end{cases}
\begin{cases}
(N_i^a)'(s) + \frac{T_i^a(s)}{R} + p(s) &= 0 \\
(T_i^a)'(s) - \frac{N_i^a(s)}{R} &= 0 \\
(M_i^a)'(s) + T_i^a(s) &= 0
\end{cases}
\tag{20}
$$

Elastic problem for the beam:

$$
\begin{cases}
\varepsilon_i^b(s) &= (u_i^b)'(s) \\
\gamma_i^b(s) &= (w_i^b)'(s) - \phi_i^b(s) \\
\chi_i^b(s) &= (\phi_i^b)'(s)
\end{cases}
\begin{cases}
N_i^b(s) &= EA_b \varepsilon_i^b(s) \\
\gamma_i^b(s) &= 0 \\
M_i^b(s) &= EJ_b \chi_i^b(s)
\end{cases}
\begin{cases}
(N_i^b)'(s) &= 0 \\
(T_i^b)'(s) &= 0 \\
(M_i^b)'(s) + T_b(s) &= 0
\end{cases}
\tag{21}
$$

Elastic problem for the diagonal:

$$
\left\{ \varepsilon_i^d(s) = (u_i^d)'(s) \right. \quad \left\{ N_i^d(s) = EA_d \varepsilon_i^d(s) \right. \quad \left\{ (N_i^d)'(s) = 0 \right.
\tag{22}
$$

The equilibrium equations are solved as a superposition of some basic solutions ($BS$) (see Figure 6) of statically determined problems.

The used basic equilibrium solutions for the arch ($BS_{ai}$), for the beam ($BS_{bi}$) and for the diagonal ($BS_{di}$) are the following:

$$
\begin{aligned}
BS_{a1}: \quad & N_a(s) = pR, & T_a(s) = 0, & \quad M_a(s) = 0. \\
BS_{a2}: \quad & N_a(s) = H\cos(\tfrac{s}{R}), & T_a(s) = H\sin(\tfrac{s}{R}), & \quad M_a(s) = HR(\cos(\tfrac{s}{R}) - \cos(\tfrac{l}{2R})). \\
BS_{a3}: \quad & N_a(s) = 0, & T_a(s) = 0, & \quad M_a(s) = m. \\
BS_{b1}: \quad & N_b(s) = H, & T_b(s) = 0, & \quad M_b(s) = 0. \\
BS_{b2}: \quad & N_b(s) = 0, & T_b(s) = 0, & \quad M_b(s) = m. \\
BS_{d1}: \quad & N_d(s) = H. & &
\end{aligned}
\tag{23}
$$

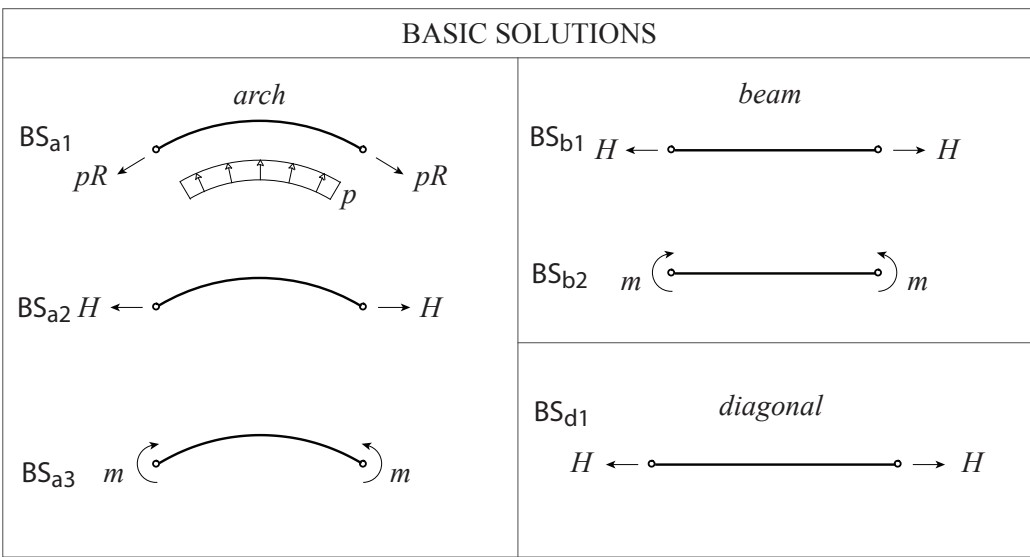

**Figure 6.** The basic solutions used in the Force Method.

### 2.3.2. Solution of the Problem (1)

In this case on the arch both external load and $X_1^a$ are applied, while on the beam only $X_1^b$ is applied, then the basic equilibrium solutions to be used for the arch are $BS_{a1}$ and $BS_{a2}$, by putting $H = -X_1^a$, while $BS_{b1}$ must be used for the beam, by putting $H = -X_1^b$ and $BS_{d1}$ for the diagonal, by putting $H = X_1$. Then, by integrating the compatibility equations for arch, beam and diagonal, we obtain the following results for the displacement fields (see Figure 7 for deformed configuration):

$$
\begin{aligned}
u_1^a(s) &= \frac{X_1 \cos(\alpha)\left(l\left(3A_b R^2 + J_b\right)\cot\left(\frac{l}{2R}\right)\sin\left(\frac{s}{R}\right) - 2s\left(A_b R^2 + J_b\right)\cos\left(\frac{s}{R}\right) - 4A_b R^2 s\cos\left(\frac{l}{2R}\right)\right)}{2A_b E J_b} \\
w_1^a(s) &= \frac{X_1 \cos(\alpha)\left(\cos\left(\frac{s}{R}\right)\left(-l\left(3A_b R^2 + J_b\right)\cot\left(\frac{l}{2R}\right) + 2A_b R^3 - 2J_b R\right) - 2s\left(A_b R^2 + J_b\right)\sin\left(\frac{s}{R}\right) + 4A_b R^3 \cos\left(\frac{l}{2R}\right)\right) + 4J_b p R^2}{2A_b E J_b} \\
\phi_1^a(s) &= \frac{2R X_1 \cos(\alpha)\left(s\cos\left(\frac{l}{2R}\right) - R\sin\left(\frac{s}{R}\right)\right)}{E J_b} \\
u_1^b(s) &= -s\frac{X_1 \sin(\alpha)}{E A_b} \\
w_1^b(s) &= \frac{4J_b p R^2 - X_1 \cos(\alpha)\csc\left(\frac{l}{2R}\right)\left(R\left(J_b - 3A_b R^2\right)\sin\left(\frac{l}{R}\right) + l\left(A_b R^2 \cos\left(\frac{l}{R}\right) + 2A_b R^2 + J_b\right)\right)}{2A_b E J_b} \\
\theta_1^b(s) &= 0 \\
u_1^d(s) &= s\frac{X_1}{E A_d}
\end{aligned}
\tag{24}
$$

Equation (19) is then solved to find the redundant reaction $X_1$.

$$
X_1 = \frac{4A_d J_b p R^2 \cos(\alpha)\sin\left(\frac{l}{2R}\right)}{A_d \cos^2(\alpha)\left(R(J_b - 3A_b R^2)\sin\left(\frac{l}{R}\right) + l\left(A_b R^2 \cos\left(\frac{l}{R}\right) + 2A_b R^2 + J_b\right)\right) + A_b J_b l_d + 2A_d J_b l \sin^2(\alpha)}
\tag{25}
$$

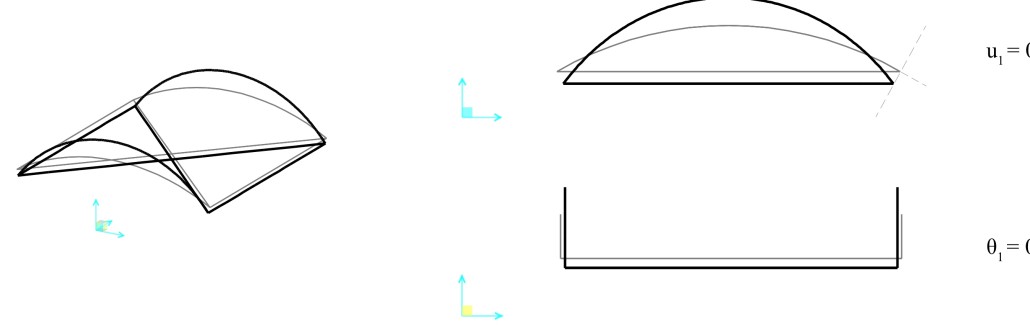

**Figure 7.** Deformed configuration of the problem (1).

### 2.3.3. Solution of the Problem (2)

In this case on the arch only $X_2^a$ is applied, while on the beam both $F$ and $X_2^b$ are applied, then the basic equilibrium solutions to be used for the arch is $BS_{a2}$, by posing $H = -X_2^a$, while $BS_{b1}$ must be used for the beam, by posing $H = F - X_2^b$ and $BS_{d1}$ for the diagonal, by posing $H = X_2$. Then, by integrating the compatibility equations for arch, beam and diagonal, we obtain the following results for the displacement fields (see Figure 8 for deformed configuration):

$$
\begin{aligned}
u_2^a(s) &= \frac{X_2 \sin(2\alpha)\csc(\alpha)\left(l\left(3A_bR^2+A_d\right)\cot\left(\frac{l}{2R}\right)\sin\left(\frac{s}{R}\right)-2s\left(A_bR^2+A_d\right)\cos\left(\frac{s}{R}\right)-4A_bR^2s\cos\left(\frac{l}{2R}\right)\right)}{4A_bEA_d} \\
w_2^a(s) &= -\frac{X_2 \sin(2\alpha)\csc(\alpha)\sin\left(\frac{s}{R}\right)\left(\cot\left(\frac{s}{R}\right)\left(l\left(3A_bR^2+A_d\right)\cot\left(\frac{l}{2R}\right)+2R\left(A_d-A_bR^2\right)\right)\right)}{4A_bEA_d} \\
&\quad +\frac{X_2 \sin(2\alpha)\csc(\alpha)\sin\left(\frac{s}{R}\right)\left(2s\left(A_bR^2+A_d\right)-4A_bR^3\cos\left(\frac{l}{2R}\right)\csc\left(\frac{s}{R}\right)\right)}{4A_bEA_d} \\
\phi_2^a(s) &= \frac{2RX_2\cos(\alpha)\left(s\cos\left(\frac{l}{2R}\right)-R\sin\left(\frac{s}{R}\right)\right)}{EA_d} \\
u_2^b(s) &= s\frac{(F-X_2\sin(\alpha))}{EA_b} \\
w_2^b(s) &= -\frac{X\cos(\alpha)\csc\left(\frac{l}{2R}\right)\left(R\left(A_d-3A_bR^2\right)\sin\left(\frac{l}{R}\right)+l\left(A_bR^2\cos\left(\frac{l}{R}\right)+2A_bR^2+A_d\right)\right)}{2A_bEA_d} \\
\theta_2^b(s) &= 0 \\
u_2^d(s) &= s\frac{X_2}{EA_d}
\end{aligned}
\tag{26}
$$

Equation (19) is then solved to find the redundant reaction $X_2$.

$$
X_2 = \frac{2A_dFA_dl\sin(\alpha)}{A_d\cos^2(\alpha)\left(R(A_d-3A_bR^2)\sin\left(\frac{l}{R}\right)+l\left(A_bR^2\cos\left(\frac{l}{R}\right)+2A_bR^2+A_d\right)\right)+A_bA_dl_d+2A_dA_dl\sin^2(\alpha)}
\tag{27}
$$

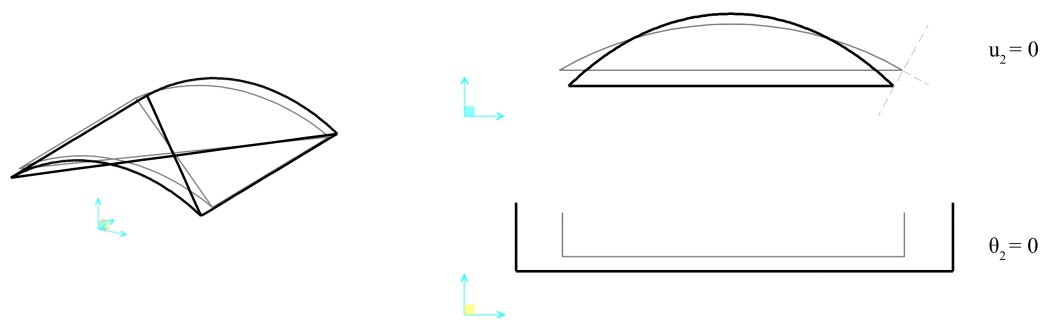

**Figure 8.** Deformed configuration of the problem (2).

### 2.3.4. Solution of the Problem (3)

In this case on the arch both $m_1$ and $X_3^a$ are applied, while only $X_3^b$ is applied on the beam, then the basic equilibrium solutions to be used for the arch are $BS_{a2}$ and $BS_{a3}$, by posing $H = -X_3^a$, and $m = m_1$, respectively, while $BS_{b1}$ must be used for the beam, by posing $H = X_3^b$ and $BS_{d1}$ for the diagonal, by posing $H = X_3$. Then, by integrating the compatibility equations for arch, beam and diagonal, we obtain the following results for the displacement fields (see Figure 9 for deformed configuration):

$$u_3^a(s) = \frac{l\sin\left(\frac{s}{R}\right)\left(X_3\cos(\alpha)\left(3A_bR^2+J_b\right)\cot\left(\frac{l}{2R}\right)+2A_bm1R\csc\left(\frac{l}{2R}\right)\right)}{2A_bEJ_b}+$$
$$\frac{-2sX_3\cos(\alpha)\left(A_bR^2+J_b\right)\cos\left(\frac{s}{R}\right)-4A_bRs\left(RX_3\cos(\alpha)\cos\left(\frac{l}{2R}\right)+m_1\right)}{2A_bEJ_b}$$

$$w_3^a(s) = \frac{X_3\cos(\alpha)\left(\cos\left(\frac{s}{R}\right)\left(-l\left(3A_bR^2+J_b\right)\cot\left(\frac{l}{2R}\right)+2A_bR^3-2J_bR\right)-2s\left(A_bR^2+J_b\right)\sin\left(\frac{s}{R}\right)+4A_bR^3\cos\left(\frac{l}{2R}\right)\right)}{EJ_b}$$
$$+\frac{2A_bm_1R\left(2R-l\csc\left(\frac{l}{2R}\right)\cos\left(\frac{s}{R}\right)\right)}{2A_bEJ_b}$$

$$\phi_3^a(s) = \frac{2RX_3\cos(\alpha)\left(s\cos\left(\frac{l}{2R}\right)-R\sin\left(\frac{s}{R}\right)\right)+2m_1s}{EJ_b}$$

$$u_3^b(s) = -s\frac{X_3\sin(\alpha)}{EA_b} \tag{28}$$

$$w_3^b(s) = \frac{2A_bm_1R\left(2R-l\cot\left(\frac{l}{2R}\right)\right)-X_3\cos(\alpha)\csc\left(\frac{l}{2R}\right)\left(l\left(2A_bR^2+J_b\right)+R\left(J_b-3A_bR^2\right)\sin\left(\frac{l}{R}\right)+A_blR^2\cos\left(\frac{l}{R}\right)\right)}{2A_bEJ_b}$$

$$\theta_b(s) = 0$$

$$u_d(s) = s\frac{X_3}{EA_d}$$

Equation (19) is then solved to find the redundant reaction $X_3$.

$$X_3 = \frac{2A_bA_dm1R\cos(\alpha)\left(2R\sin\left(\frac{l}{2R}\right)-l\cos\left(\frac{l}{2R}\right)\right)}{A_d\cos^2(\alpha)\left(R(J_b-3A_bR^2)\sin\left(\frac{l}{R}\right)+l\left(A_bR^2\cos\left(\frac{l}{R}\right)+2A_bR^2+J_b\right)\right)+A_bJ_bl_d+2A_dJ_bl\sin^2(\alpha)} \tag{29}$$

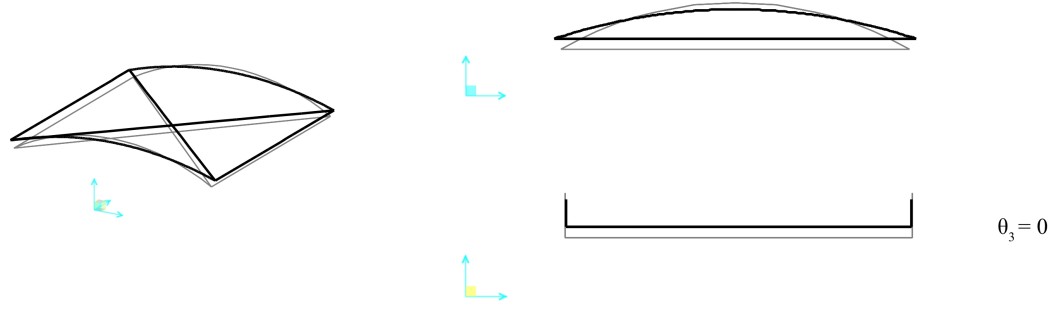

$\theta_3 = 0$

**Figure 9.** Deformed configuration of the problem (3).

2.3.5. Solution of the Problem (4)

$$\begin{aligned}
u_i^a(s) &= 0 \\
w_i^a(s) &= 0 \\
\phi_i^a(s) &= 0 \\
u_i^b(s) &= 0 \\
w_i^b(s) &= -\frac{m_2\left(L^2-4s^2\right)}{4EJ_b} \\
\theta_i^b(s) &= \frac{2m_2s}{EJ_b} \\
u_i^d(s) &= 0
\end{aligned} \tag{30}$$

In this last case the redundant reaction $X_4$ vanishes (see Figure 10 for deformed configuration).

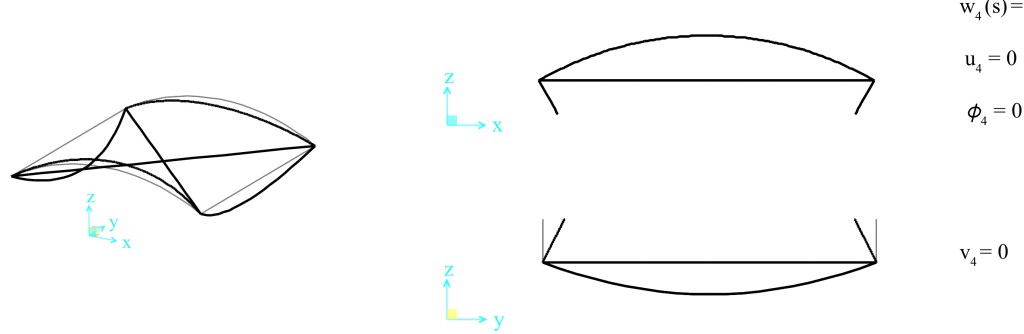

**Figure 10.** Deformed configuration of the problem (4).

### 2.4. The Identified Constitutive Coefficients

Once the four fine problems are solved, it is possible to put solutions in Equations (12)–(18) by means of the following substitutions:

$$\hat{w}_1(s) = \frac{w_1^a(s)}{pR}, \hat{w}_2(s) = \frac{w_2^a(s)}{F}, \hat{v}_1 = \frac{v_1^a(s)}{pR}, \hat{v}_2 = \frac{v_2^a(s)}{F}, \hat{w}_3 = \frac{w_3^a(s)}{m_1} \tag{31}$$

$$\hat{\phi}_1 = \frac{\phi_1^a(s)}{pR}, \hat{u}_3 = \frac{u_3^a(s)}{m_1}, \hat{\phi}_3 = \frac{\phi_3^a(s)}{m_1}, \hat{\theta}_4 = \frac{\theta_4^a(s)}{m_2} \tag{32}$$

Then, by substituting $\beta = \frac{l}{2R}$, it is possible to write all geometrical parameters as functions of $\beta$:

$$\alpha = \tan^{-1}\left(\frac{\beta}{\sin(\beta)}\right) \tag{33}$$

$$l_d = l\sqrt{1 + \left(\frac{\sin(\beta)}{\beta}\right)^2} \tag{34}$$

Finally, the identified constitutive coefficients, as functions of the angle $\beta$ can be represented as follows:

$$D_{11}(\beta) = \frac{l}{EA_b}\left(1 + \frac{8(\sin\beta/\beta)^2 A_d J_b}{c_1(\beta)l^2 A_b A_d - c_2(\beta)A_b J_b - c_3(\beta)A_d J_b}\right) \tag{35}$$

$$D_{12}(\beta) = \frac{l}{EA_b}\left(\frac{8A_d J_b}{c_1(\beta)l^2 A_b A_d - c_2(\beta)A_b J_b - c_3(\beta)A_d J_b}\right)$$

$$D_{22}(\beta) = \frac{l}{E\left(\left(1 + \left(\frac{\sin\beta}{\beta}\right)^2\right)^{\frac{3}{2}}A_b + 2A_d\right)}\left(\left(1 + \left(\frac{\sin\beta}{\beta}\right)^2\right)^{\frac{3}{2}} + \frac{2\left(c1(\beta)l^2 A_b - 4\left(1 + \frac{\sin 2\beta}{2\beta}\right)J_b\right)A_d^2}{c_1(\beta)l^2 A_b A_d - c_2(\beta)A_b J_b - c_3(\beta)A_d J_b}\right)$$

$$D_{13}(\beta) = \frac{l^2}{EJ_b}\left(\frac{4\left(\frac{\sin\beta}{\beta}\right)^2(1 - \beta\cot\beta)}{\beta}\frac{A_d J_b}{c_1(\beta)l^2 A_b A_d - c_2(\beta)A_b J_b - c_3(\beta)A_d J_b}\right)$$

$$D_{23}(\beta) = \frac{l^2}{EJ_b}\left(\frac{4(1 - \beta\cot\beta)}{\beta}\frac{A_d J_b}{c_1(\beta)l^2 A_b A_d - c_2(\beta)A_b J_b - c_3(\beta)A_d J_b}\right)$$

$$D_{33}(\beta) = \frac{l}{EJ_b}\left(1 + \frac{2(\cos\beta - \sin\beta/\beta)^2}{\beta^2}\frac{l^2 A_d A_b}{c_1(\beta)l^2 A_b A_d - c_2(\beta)A_b J_b - c_3(\beta)A_d J_b}\right)$$

$$D_{44}(\beta) = \frac{l}{EJ_b}$$

where the dimensionless coefficients $c_1(\beta), c_2(\beta), c_3(\beta)$ are defined as follows:

$$
\begin{aligned}
c_1(\beta) &= \frac{1}{\beta^2}\left(\frac{3\sin 2\beta}{2\beta} - (2+\cos 2\beta)\right) \\
c_2(\beta) &= 4\left(\frac{\beta}{\sin\beta}\right)^2\left(1+\left(\frac{\sin\beta}{\beta}\right)^2\right)^{\frac{3}{2}} \\
c_3(\beta) &= 8\left(\frac{\beta}{\sin\beta}\right)^2 + 4\left(1+\frac{\sin 2\beta}{2\beta}\right).
\end{aligned}
\tag{36}
$$

It easy to show that:

$$
\lim_{\beta\to 0} c_1(\beta) = 0, \ \lim_{\beta\to 0} c_2(\beta) = 8\sqrt{2}, \ \lim_{\beta\to 0} c_3(\beta) = 16
$$

Consequently, taking the limit of the identified coefficients for small angles $\beta$, we obtain the same coefficients introduced in [6]:

$$
\begin{aligned}
\overline{D}_{11} &= \lim_{\beta\to 0} D_{11}(\beta) &= \frac{l}{EA_b}\left(\frac{\sqrt{2}A_b+A_d}{\sqrt{2}A_b+2A_d}\right) & \qquad \overline{D}_{13} &= \lim_{\beta\to 0} D_{13}(0) = 0 \qquad (37) \\
\overline{D}_{12} &= \lim_{\beta\to 0} D_{12}(0) &= -\frac{l}{EA_b}\left(\frac{8A_dJ_b}{\sqrt{2}A_b+2A_d}\right) & \qquad \overline{D}_{23} &= \lim_{\beta\to 0} D_{23}(0) = 0 \\
\overline{D}_{22} &= \lim_{\beta\to 0} D_{22}(0) &= \frac{l}{EA_b}\left(\frac{\sqrt{2}A_b+A_d}{\sqrt{2}A_b+2A_d}\right) & \qquad \overline{D}_{33} &= \lim_{\beta\to 0} D_{33}(0) = \frac{l}{EJ_b}
\end{aligned}
$$

Figure 11 shows the behaviour of the identified coefficients as functions of $\beta$, superimposed with the value obtained in [6]. It is worth noting that:

- For $\beta = 0$ (flat REV) the values coincide with that of [6].
- For $\beta = 0$ there is no coupling between membrane and flexural behaviour, i.e., $D_{13} = D_{23} = 0$.
- For $\beta > 0$ (curved REV) the membrane coefficients $D_{11}$ and $D_{22}$ assume positive values increasing monotonically with $\beta$, while the mixed membrane coefficient $D_{12}$ (Poisson effect) assume negative values whose absolute value decreases with $\beta$. Then the overall membrane stiffness increases with the curvature of the REV.
- For $\beta > 0$ (curved REV) the coupling coefficients $D_{13} = D_{23} = 0$, are non monotonic functions of $\beta$, then there exists a value of $\beta$ giving the maximum coupling.
- The flexural coefficient $D_{44}$ is not affected by $\beta$ because we assumed a geometry curved only in $x$ direction.

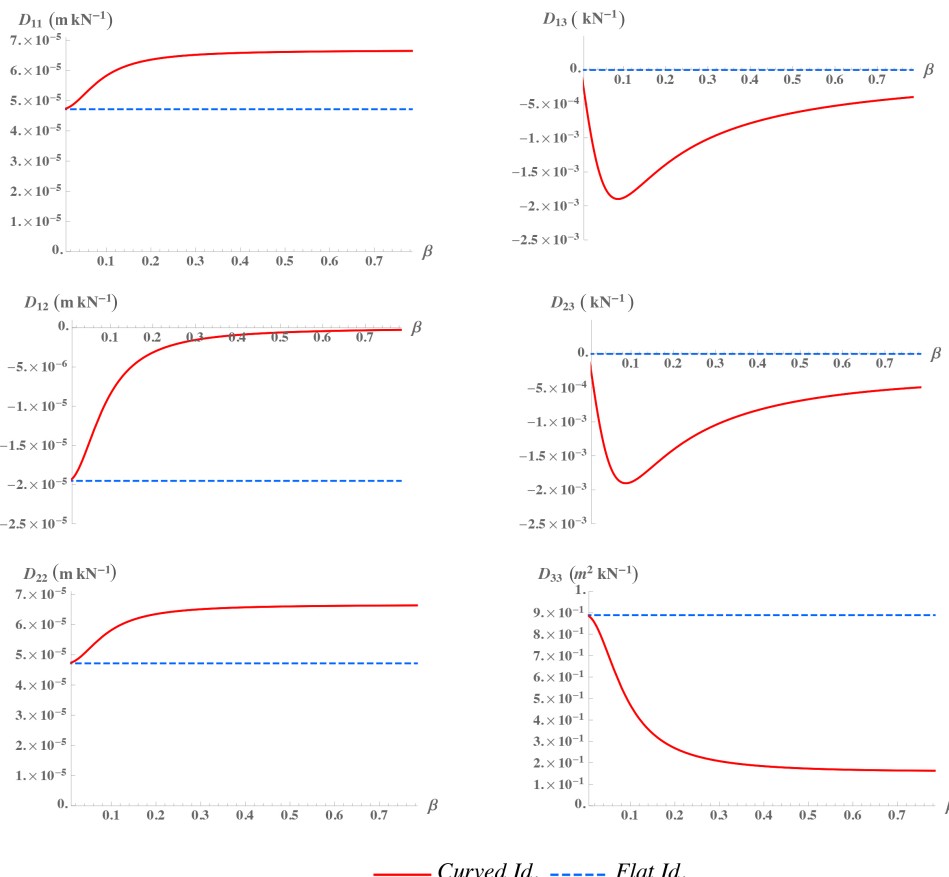

Figure 11. The six curves of $D_{11}(\beta), D_{12}(\beta), D_{22}(\beta), D_{13}(\beta), D_{23}(\beta), D_{33}(\beta)$, (red, solid line), superimposed with the corresponding $\overline{D}_{11}, \overline{D}_{12}, \overline{D}_{22}, \overline{D}_{13}, \overline{D}_{23}, \overline{D}_{33}$, (blue, dashed line), in the case of $l = 1.0$ m, $E = 1.0 \times 10^4$ MPa, $A_b = 1.5 \times 10^{-3}$ m$^2$, $Ad = 1.5 \times 10^{-3}$ m$^2$, $Jb = 1.35 \times 10^{-6}$ m$^4$.

## 3. Results

### 3.1. Homogeneous Load Case

In order to validate the identification procedure, the response of coarse model is numerically compared to that of the fine one.

The numerical analyses are carried out with the finite element software SAP2000 [18]. A cylindrical gridshell having a 1.90 m radius is chosen as fine model, divided in 36 modules (six longitudinal and six circumferential, see Figure 12). The characteristic length of each module is 1.00 m. All structural elements are modelled as timber laths having a rectangular cross-section, 50 mm wide and 30 mm deep; the chosen material belongs to the strength class D30 [19], with the mean modulus of elasticity parallel to the fibers equal to 11,000 MPa.

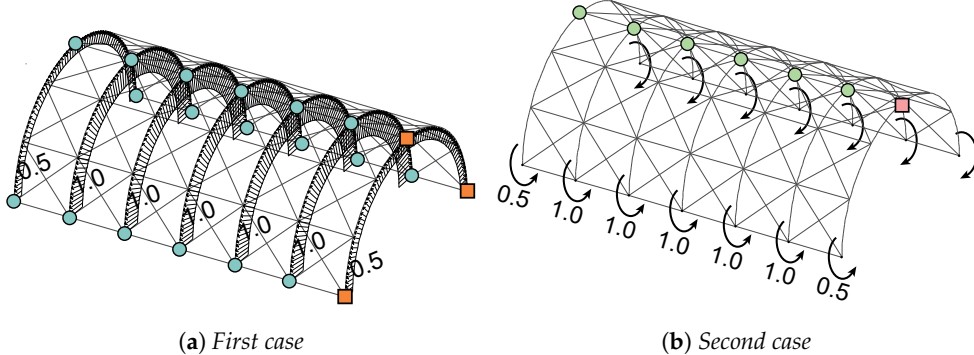

(**a**) *First case*　　　　　　　　　　(**b**) *Second case*

**Figure 12.** Load and boundary conditions.

The fine model has a global orientation as defined in Figure 1. It is analysed under two homogeneous load conditions: a uniform pressure load distributed on the arches (Figure 12a) and a distributed moment about the axis of the straight edge beams (Figure 12b).

Boundary conditions (*b.c.*) are assigned with reference to the displacement components defined in Section 2.4. In both cases of Figure 12a,b all the nodes of the gridshell depicted as a circle have $b.c. := \{u, \phi, \theta\} = 0 \cup \{v, w\} \neq 0$. However, in order to avoid the rigid translation in $y$ direction, the arch where nodes are depicted with squares have $b.c. := \{u, v, \phi, \theta\} = 0 \cup \{w\} \neq 0$.

Coarse model is analysed under equivalent load and boundary conditions; the results in terms of deformed shapes for both models are shown in Figure 13.

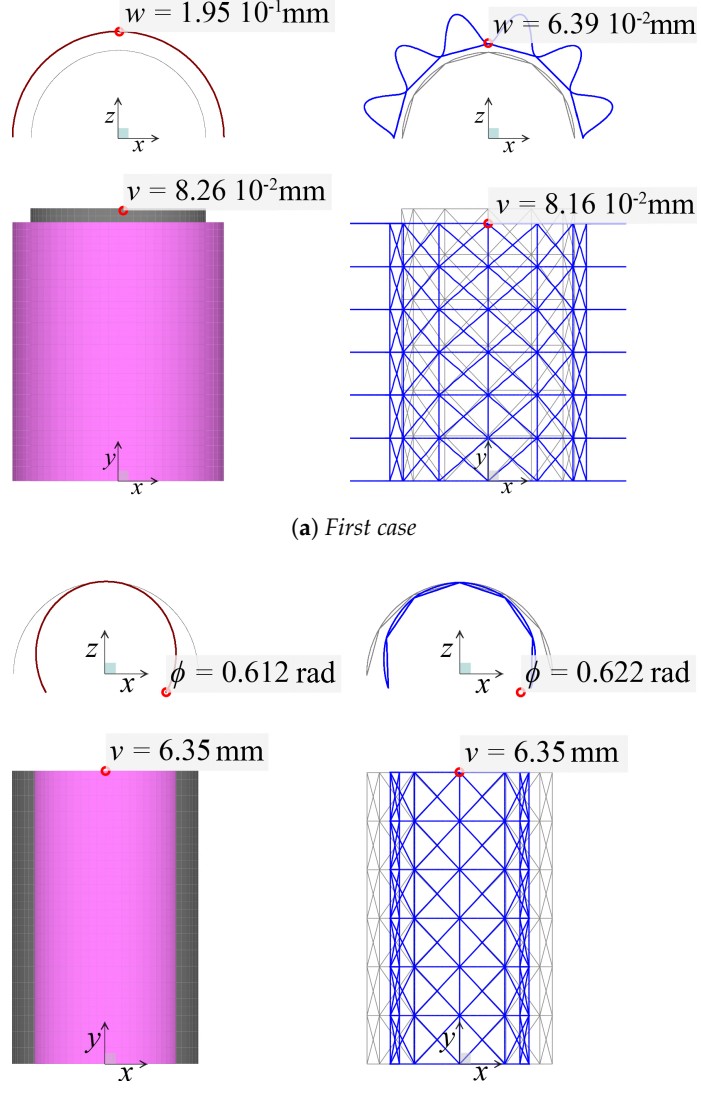

(**a**) *First case*

(**b**) *Second case*

**Figure 13.** Deformed shapes.

In the first case (a), the coarse model behavior can be characterized just by $\epsilon_1$ and $\epsilon_2$. These strain measures are constant all along whole shell; in particular, $\epsilon_1 = 1.02 \times 10^{-4}$ and $\epsilon_2 = -1.38 \times 10^{-5}$. By applying Equation (7), the $w$ component of the displacement vector is calculated equal to $6.4 \times 10^{-2}$ m. Thus, the percent error between the coarse model and the fine model is below 1%.

In the second case (b) instead, the strain measures $\chi_1$, $\epsilon_1$ and $\epsilon_2$ characterize the coarse model behavior, and their numerical values are calculated: $\chi_1 = 0.20 \, \text{m}^{-1}$, $\epsilon_1 = -1.03 \times 10^{-3}$, $\epsilon_2 = -1.06 \times 10^{-3}$. Applying Equation (7) again in order to calculate the $\phi$ component of the

rotation vector, this is evaluated as 0.102 rad. However, the obtained value of $\phi$ corresponds to the rotation of the arches' end sections of one module.

Being $\chi_1$ constant in the problem of Figure 12b it can be deducted that rotation $\phi$ is linear. Thus, $\phi = 0.612$ rad at the end section of the whole arch of the fine model when composed of six modules. By the numerical analysis of the fine model, $\phi = 0.622$ at the end section of the arches; therefore, the error of the coarse model is below to 2%.

The $v$ component of the displacement vector can be verified in the same way for both load cases; here, the error of the coarse model is again negligible.

### 3.2. Influence of the Curvature: A Non–Homogeneous Example

A gridshell can be analysed using continuum models obtained by different identifications. In this section, the focus is on the comparison of two continua obtained by constitutive identifications based on different REVs. The continuum model proposed in the present is now compared to that defined in [14]; for the sake of simplicity, the first one is named continuum A (CA) and the second one continuum B (CB).

The responses of the different models are compared in terms of mechanical work ($W$), numerically evaluated as the work spent by the external forces over the displacements.

A first comparison (shown in Figure 14) is done basing on the results of five tests, characterized by the same radius (R) but a different number of modules ($n$); in particular, the radius is 20 m and the number of the modules ranges from $12 \times 12$ to $36 \times 36$. Note that in this case $\beta$ is fixed, having $l$ and $R$ fixed values.

Each model, namely the fine and the coarse, is analyzed under the same uniform vertical load equal to $1.0 \, \text{N/m}^2$. The shell is supported along the boundary straight edges by pins on one side and by rollers on the other side.

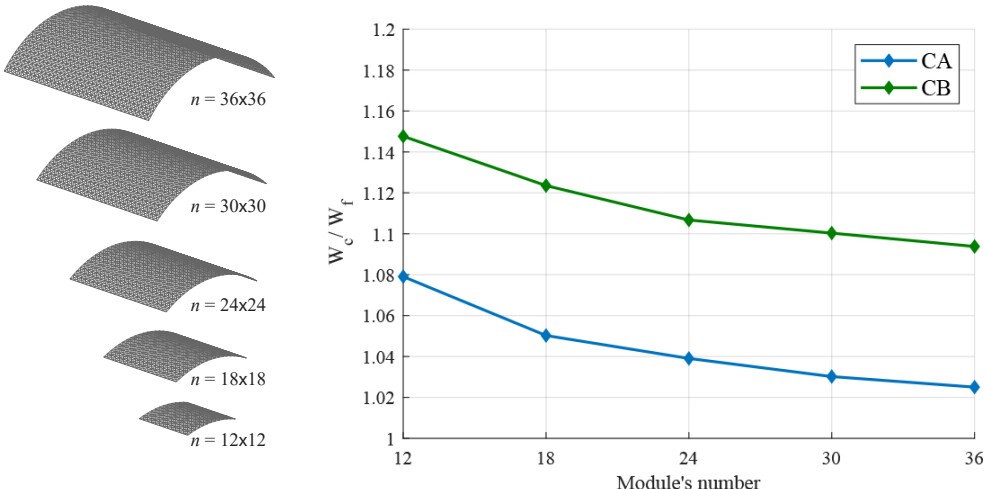

**Figure 14.** Percent error of CA and CB from the fine model with respect to modules' number.

Figure 14 depicts the ratio ($W_c/W_f$) of the mechanical work of the continuum model ($W_c$) over the mechanical work of the fine model ($W_f$). As could be expected, when the number of modules increases the response of both continua tend to that of the corresponding fine model. Although the trends are similar, the response of CA is closer to that of the fine model than CB's one. Since both curves take values greater than 1, the equivalent continua reveal to be more flexible than the corresponding gridshells.

A second comparison is performed in order to study the effect of the geometric curvature $\kappa = 1/R$. Seven numerical tests are performed on purpose, where the gridshells are subject to the same loads and b.c. as in the previous test. This time the number of modules is fixed to ($n = 24$), while the curvature $\kappa$ ranges from 0 to $0.1125 \, \text{m}^{-1}$ (Figure 15).

When $\kappa$ is zero, i.e., flat geometry, the response of the two continua clearly provides the same error compared to the fine model. However, interestingly the responses of the two continua rapidly diverge as the curvature increases, as Figure 15 shows.

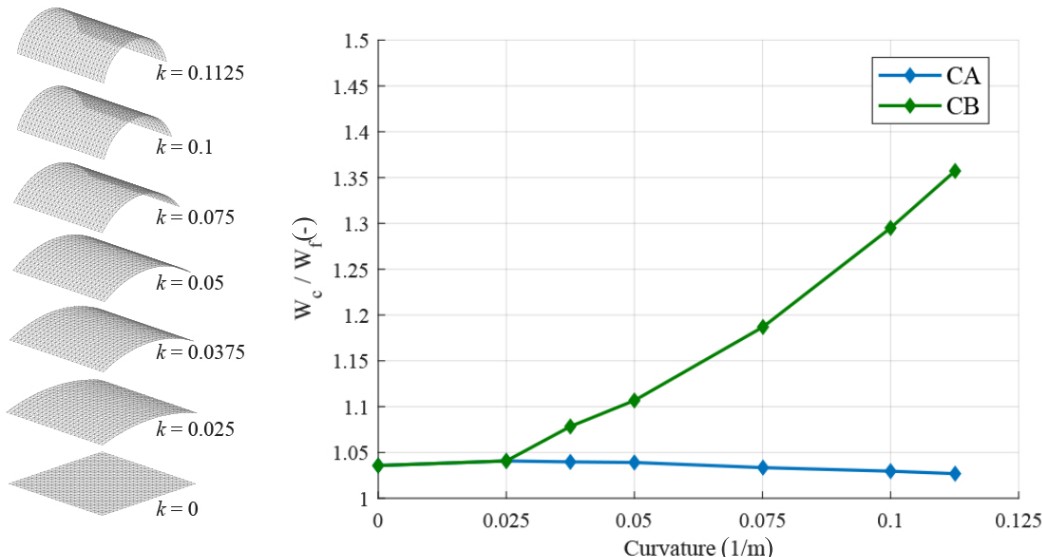

**Figure 15.** Percent error of CA and CB from the fine model with respect to the curvature $\kappa = 1/R$. Being $l = 1$ m, in this case $\kappa = 2\beta$. $\kappa \in [0, 0.1125]$ m$^{-1}$.

## 4. Conclusions

In this work, we demonstrated that a proper constitutive identification of curved gridshells leads to the expected coupling of membrane and flexural behavior for an equivalent continuum model. This result is due to a novel approach taking into account the geometric curvature in the equivalent continuum model. Even the geometrically simplest case of a barrel gridshell shows how much the solution can diverge while adopting an equivalent continuum derived from a flat REV.

As highlighted in the introduction, the transition from the fine model to the continuum one is a useful tool in a phase of generation and selection of the global form of gridshells, when a precise definition of the grid is probably useless and cumbersome. Therefore, the constitutive identification adopted in this work can be a useful tool for the preliminary design stage. Moreover, this work can be considered a first step towards a methodologically correct approach aiming at defining equivalent continua of gridshells with double curvature.

**Author Contributions:** Conceptualization, M.L.R., S.G., V.V. and G.S.; Methodology, M.L.R., S.G., V.V. and G.S.; Implementation, M.L.R., S.G., V.V. and G.S.; Writing, M.L.R., S.G., V.V. and G.S.; Supervision G.S. All authors have read and agreed to the published version of the manuscript.

**Funding:** This research received no external funding.

**Institutional Review Board Statement:** Not applicable.

**Informed Consent Statement:** Not applicable.

**Data Availability Statement:** Not applicable.

**Conflicts of Interest:** The authors declare no conflict of interest.

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
