# Peer review of "Equivalent Shell Model of Elastic Gridshells Including the Effect of the Geometric Curvature"

_2673-3161, doi:10.3390/applmech2030036_

Round 1

Reviewer 1 Report

Dear the Editor in Chief,

Attached file concernes the

Review of the paper [Applied Mechanics] Manuscript ID: applmech-1339901 - Review.

Reviewer 2 Report

The paper “Equivalent shell model of elastic gridshells including the effect of the geometric curvature” is well written and can be read easily by anyone familiar with the beam and shell theories. The research field of the manuscript is the object of the journal.

In my opinion, some equations in this work don’t satisfy the equilibrium conditions. I have listed these equations as an attached file. Therefore, I agree with publication of this manuscript in “Applied Mechanics”, on the understanding that the authors will elaborate the manuscript complying with a major correction according to the attached file “REVIEW-Equivalent shell model.pdf”.

Reviewer 3 Report

The aim of the manuscript under report is to provide a proper definition of the equivalent stiffness describing the transition from the fine to the coarse model, that is, from the gridshell model to that of the shell.

In order to determine the elastic coefficients of the continuum model equivalent to the gridshell, a continualization procedure of a periodic discrete system is used, which follows the lines of the homogenization methods of heterogeneous continuous models.

The discrete and the continuum models are described. The solutions of these models are obtained by solving complicate systems of equations. It seems that the calculations are correct.

Nice pictures of some models are plotted.

In my opinion this paper has a high degree of originality, it is clearly presented and supported by examples.

On basis of the above arguments, I recommend the publication.

Round 2

Reviewer 1 Report

I thank the authors  for your contribution.

Sincerely.

Reviewer 2 Report

In my first review, I gave a hint to improve the derivation. The authors ignored the instructions and only adapted Fig. 4. The element in its current form in Fig. 4 can only be used for special cases of real models. But, I would accept it.